# Analysis of the Oscillating Motion of a Solid Body on Vibrating Bearers

**Kuatbay Bissembayev [1,2], Assylbek Jomartov [1,*], Amandyk Tuleshov [1] and Tolegen Dikambay [2]**

[1] Ministry of Education and Science of Kazakhstan, Institute Mechanics and Mechanical Engineering, Almaty 050010, Kazakhstan

[2] Ministry of Education and Science of Kazakhstan, Abai Kazakh National Pedagogical University, Almaty 050000, Kazakhstan

\* Correspondence: legsert@mail.ru; Tel.: +7-777-329-5999

**Abstract:** This article considers the oscillation of a solid body on kinematic foundations, the main elements of which are rolling bearers bounded by high-order surfaces of rotation at horizontal displacement of the foundation. Equations of motion of the vibro-protected body have been obtained. It is ascertained that the obtained equations of motion are highly nonlinear differential equations. Stationary and transitional modes of the oscillatory process of the system have been investigated. It is determined that several stationary regimes of the oscillatory process exist. Equations of motion have been investigated also by quantitative methods. In this paper the cumulative curves in the phase plane are plotted, a qualitative analysis for singular points and a study of them for stability are performed. In the Hayashi plane a cumulative curve of a body protected against vibration forms a closed path which does not tend to the stability of a singular point. This means that the vibration amplitude of a body protected against vibration does not remain constant in a steady state, but changes periodically.

**Keywords:** vibroprotection; seismic; rolling bearer; vibration; non-linear vibrations; cumulative curves; singular point

## 1. Introduction

The issue of vibration protection for devices and equipment is one of the main directions of development of the theory for vibrations of mechanical systems.

The theory of nonlinear vibration isolation has witnessed significant developments because of pressing demands for the protection of structural installations, nuclear reactors, mechanical components, and sensitive instruments from earthquake ground motion, shocks, and impact loads. In views of these demands, engineers and physicists have developed different types of nonlinear vibration isolators. This article [1] presents a comprehensive assessment of the recent developments of nonlinear isolators in the absence of active control means. It does not deal with other means of linear or nonlinear vibration absorbers. The article is closed by conclusions, which highlight resolved and unresolved problems and recommendations for future research directions.

A new, passive, vibro-protective device of the rolling-pendulum tuned mass damper type is presented [2] that, relying on a proper three-dimensional guiding surface, can simultaneously control the response of the supporting structure in two mutually orthogonal horizontal directions. Unlike existing examples of ball vibration absorbers, mounted on spherical recesses and effective for axial-symmetrical structures, the new device is bidirectional tunable, by virtue of the optimum shape of the rolling cavity, to both fundamental structural modes, even when the corresponding natural frequencies are different, in such a case recurring to an innovative non-axial-symmetrical rolling guide.

A new optimization method for a tuned mass damper (TMD) system is proposed in the paper [3], based on the artificial fish swarm algorithm (AFSA), and the primary structural damping is taken into consideration. The optimization goal is to minimize the maximum dynamic amplification factor of the primary structure under external harmonic excitations.

The paper [4] deals with the performance analysis of a vibration-isolation system for Michelangelo Buonarroti's famous Ronadanini Pietà statue based on the monitoring and analysis of vibration signals. A tuned mass-damper-inerter is introduced in order to increase the effectiveness of the isolator in horizontal direction. Specifically, a multi-degree-of-freedom (MDOF) model for the system, including non-linear terms, is proposed.

Creating vibration protection devices, using rolling bearings, is currently widespread in transport techniques to prevent transported oversize cargoes from longitudinal accelerations, for the seismic protection of structures and in other areas of modern technology. However, further progress in improving vibro-protective rolling bearings necessitates dynamic properties research and finding more advanced design solutions on the basis of this research. Most modern vibro-supporting devices use movable supports, bounded by spherical surfaces.

Articles [5,6] are focused on the technical issues, as well as on issues of improving the engineering calculation for kinematic foundations designed by this author.

Work by Y.D. Cherepinskiy [7] considers the motion of structures on the kinematic piers of a particular design, proposed by the author. It investigates a motion, originating on a plane without rolling friction, which has a significant impact on the character of the system motion.

The passive neutralization oscillations systems for high-rise construction are under consideration [8]. Their advantages and disadvantages have been revealed. A roller oscillation neutralization system for high-rise constructions subject to seismic affecting is offered. The principle of its work is described and its advantages are estimated. A mathematical movement model for carrying and carried bodies is made. Low-frequency oscillation vibration protection systems under the influence of external harmonious impact are considered. Optimum adjustment parameters for a roller damper in the structure of the compensation system are defined.

The nonlinear normal vibration modes of a mechanical system having the pendulum vibration absorber are considered [9]. The coupled and localized vibration modes are selected. In the last case the main vibration energy is concentrated in the pendulum, so this vibration mode is the most appropriate for the vibration absorption. The modes stability is investigated.

The work [10] researches low-frequency vibrations of vibro-protective system of solid bodies formed by a roller damper and a moving load-carrying body under the action of external harmonic excitations. The dynamic equations of combined motion of the working body of the damper over the hinged roller without sliding and the load-carrying body are deduced and numerically analyzed. A new procedure for evaluation of the optimal parameters of adjustment of roller dampers in nonlinear systems is proposed.

Longitudinal vibrations are investigated for the four-mass vibration-resistant system of the following solid bodies: long cargo, turnstile with roller shock absorbers, and the coupling of two flat cars after their collision with a braked hammer car [11]. The level of dynamic loads applied to the elements of the vibration-resistant system is numerically analyzed.

Low-frequency vibrations of a vibro-protection "roller damper-movable bearing body" system of rigid bodies under the action of an external harmonic excitation are considered. The working surface of the damper working body is formed by a brachistochrone. The dynamic equations of common no-slip motion of the damper working body on a hinged roller and of the bearing body are formulated. The roller damper tuning parameters are determined [12].

In all these studies, to get the final results we considered a vibration device, the bearing elements of which are bounded by spherical surfaces. A common disadvantage of all these devices is the lack of reliability at a high level of seismic disturbance. Practically, such systems behave linearly with

respect to disturbance and, to suppress the vibrations of the protected bodies, such systems of seismic protection are complemented by special devices of dry friction with the specified backlash.

This work [13] studies the features of vibration motion of an orthogonal mechanism with disturbances, such as restricted power in the presence of a fixed load on the horizontal link. Dynamic and mathematical models were prepared, and the operating conditions' fields of existence for the vibration mechanism in terms of driving power were defined.

In the work [14] the mathematical expressions for the rolling resistance arising from rolling of a bearing, bounded by high order rotational surfaces are obtained.

The work [15] contains a systematic depiction of non-linear systems analysis methods, described by differential equations of second-rate. This work also contains topological and graphical methods, applicable for the calculation of autonomic and, especially, non-autonomic systems.

In the book [16] the theory of non-linear vibrations is expounded, the topic of great interest at present because of its many applications to important fields in physics and engineering.

This paper [17] presents the results of modeling of vibrations of a rigid rotor caused by the degradation of hydrodynamic bearings. The model is composed by applying equations of nonlinear hydrodynamic forces and the measured parameters of a real rotary machine.

The work contains geometrical non-linear analysis derived according to the Hamilton principle.

In the work [18] a systematic method is developed for the dynamic analysis of structures with sliding isolation, which is a highly non-linear dynamic problem. According to the proposed method, a unified motion equation can be adapted for both stick and slip modes of the system. Unlike the traditional methods by which the integration interval has to be chopped into infinitesimal pieces during the transition of sliding and non-sliding modes, the integration interval remains constant throughout the whole process of the dynamic analysis by the proposed method so that the accuracy and efficiency in the analysis of the non-linear system can be enhanced to a large extent.

The effects of neglecting small harmonic terms in the estimation of the dynamic stability of the steady state solution determined in the frequency domain are considered in the paper [19]. For that purpose, a simple single-degree-of-freedom piecewise linear system excited by a harmonic excitation is analyzed. In the time domain, steady state solutions are obtained by using the method of piecing the exact solutions (MPES) and in the frequency domain, by the incremental harmonic balance method (IHBM). The stability of the solutions obtained in the frequency domain by IHBM is determined by using the Floquet-Liapounov theorem and by digital simulation of the corresponding disturbed motion.

The aim of the present work is to study the dynamics of vibro-protection systems, the main elements of which are rolling bearings, bounded by surfaces of rotation of high order (in the absence of rolling friction).

## 2. Statement of the Problem

Under the influence of longstanding loads, surfaces of a rolling bearing and bases change their curvature. There are two cases: under the effect of longstanding loads, the curvature radius of the rolling supports surface and bases changes by a finite amount and forms a finite area of support. From an analytical point of view, the dynamic properties of these supports are close to the dynamic properties of the rolling support with bounded surfaces of a high-order.

Let us consider the principle of work of the kinematic foundation of moving supporting elements, which is the rolling bearing with bounded surfaces of rotation of a high ($n$, $m$) order (Figure 1).

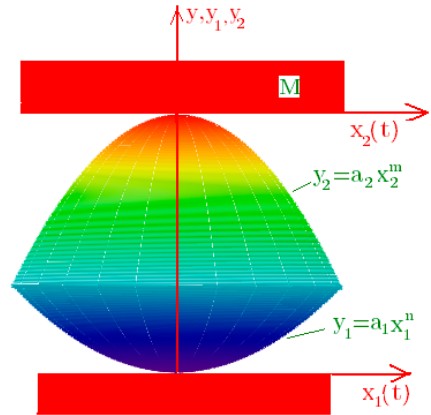

**Figure 1.** Scheme of rolling bearers with bearing.

Figure 1 demonstrates the rolling bearing (the object I) with bounded (top and bottom) surfaces of rotation, expressed by formulas:

$$\widetilde{y}_1 = \widetilde{a}_1 \widetilde{x}_1^n, \quad \widetilde{y}_2 = \widetilde{a}_2 \widetilde{x}_2^m \tag{1}$$

having a common axis of symmetry; but objects 2 and 3 are a stationary base (foundation) and the inner coat of a vibro-protected body. The specifics of such support is that the radius of curvature in the vicinity of the central support points tends to infinity and decreases with increasing distance from the axis of symmetry, i.e., there is straightening of the bearing surfaces in the vicinity of the central point. When considering $n$ to infinity ($n \to \infty$), the rolling bearing I shall take a cylindrical shape.

In the systems, the restoring force arises because of the increase of potential energy when picking up the support's centre of gravity or supports and protected body. Contact with the rolling bearing surfaces of the vibro-insulated body and the foundation will be assumed as planes.

We assume that the foundation of the considered body has a small plane length, allowing us not to take into account the asynchrony of transmission of external influence from the various points of the foundation and vibro-insulated body. Equation (1) refers to the coordinate system associated with the rolling bearings (see Figure 2). The curvature radius of the vertices of these surfaces at $n, m > 2$ tends to infinity, i.e., there is straightening of the bearing surfaces. Let us denote the horizontal offset of the bases as $\widetilde{x}_0(t)$. As $\widetilde{x}(t)$ we denote a displacement of the upper body, supporting on the rolling bearing.

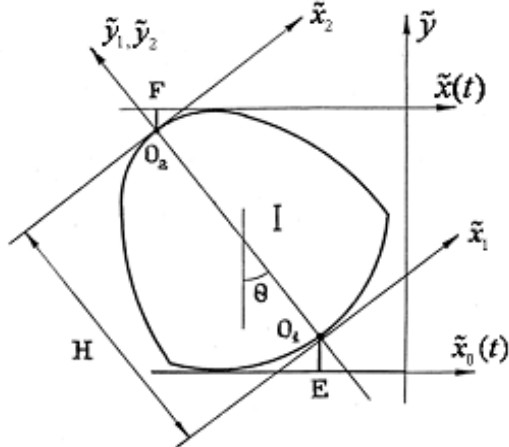

**Figure 2.** Scheme of coordinates of rolling bearing surfaces of high order.

In Figure 2 the rolling bearing is shown in the position when the base and body are offset relative to each other by $(\widetilde{x} - \widetilde{x}_0)$.

We define the dependence between the horizontal relative displacement of the foundation of a resilient construction on rolling bearers and their vertical shift. Let us introduce a new notation (Figure 2):

$$\theta = \frac{\widetilde{x} - \widetilde{x}_0}{H}$$

where $H$ and $\theta$ are the height and angle of the rotational displacement, respectively.

On the other hand

$$\theta = \frac{d\widetilde{y}_1}{d\widetilde{x}_1} = \frac{d\widetilde{y}_2}{d\widetilde{x}_2}, \quad \theta = n\widetilde{a}_1\widetilde{x}_1^{n-1} = m\widetilde{a}_2\widetilde{x}_2^{m-1} \tag{2}$$

Through Equation (2), we express $\widetilde{x}_1$, $\widetilde{y}_1$ $and$ $\widetilde{x}_2$, $\widetilde{y}_2$ via $\theta$ as

$$\widetilde{x}_1 = \frac{\theta^{\frac{1}{n-1}}}{(n\widetilde{a}_1)^{\frac{1}{n-1}}}, \quad \widetilde{y}_1 = \frac{\widetilde{a}_1}{(n\widetilde{a}_1)^{\frac{1}{n-1}}}\theta^{\frac{n}{n-1}}, \quad \widetilde{x}_2 = \frac{\theta^{\frac{1}{m-1}}}{(m\widetilde{a}_2)^{\frac{1}{m-1}}}, \quad \widetilde{y}_2 = \frac{\widetilde{a}_2}{(m\widetilde{a}_2)^{\frac{1}{m-1}}}\theta^{\frac{m}{m-1}} \tag{3}$$

Vertical displacement of the foundation is written as:

$$\widetilde{y} = H\cos\theta = H + O_1E + O_2F, \quad \widetilde{y} = -2H\sin^2\frac{\theta}{2} = H + O_1E + O_2F \tag{4}$$

where

$$O_1E = \widetilde{x}_1\sin\theta - \widetilde{y}_1\cos\theta, \quad O_2F = \widetilde{x}_2\sin\theta - \widetilde{y}_2\cos\theta$$

Transforming the function $\overline{y}$ to the range of Taylor and taking into account the first order of smallness of the angle $\theta$, for relation Equation (4) we get the expression:

$$\widetilde{y} = -H\frac{\theta^2}{2} + \widetilde{x}_1\theta - \widetilde{y}_1 + \widetilde{x}_2\theta - \widetilde{y}_2 \tag{5}$$

Substituting expression Equations (3)–(5), we get

$$\widetilde{y} = -H\frac{\theta^2}{2} + \frac{(n-1)\widetilde{a}_1}{(n\widetilde{a}_1)^{\frac{n}{n-1}}}\theta^{\frac{n}{n-1}} + \frac{(m-1)\widetilde{a}_2}{(m\widetilde{a}_1)^{\frac{m}{m-1}}}\theta^{\frac{m}{m-1}} \tag{6}$$

Taking into account the expression Equation (2), we rewrite the expression Equation (6) in the form

$$\widetilde{y} = -\frac{1}{2H}(\widetilde{x} - \widetilde{x}_0)^2 + \frac{(n-1)\widetilde{a}_1}{(Hn\widetilde{a}_1)^{\frac{n}{n-1}}}(\widetilde{x} - \widetilde{x}_0)^{\frac{n}{n-1}} + \frac{(m-1)\widetilde{a}_2}{(Hm\widetilde{a}_1)^{\frac{m}{m-1}}}(\widetilde{x} - \widetilde{x}_0)^{\frac{m}{m-1}} \tag{7}$$

Term Equation (7) defines the dependence between the horizontal relative movements of the bodies' bases on the rolling bearings with straightened surfaces and their vertical displacements. In the case, when $n = m$, term Equation (7) takes the form

$$\widetilde{y} = -\frac{1}{2H}(\widetilde{x} - \widetilde{x}_0)^2 + \frac{(n-1)}{(Hn)^{\frac{n}{n-1}}}\left(\frac{1}{\sqrt[n-1]{\widetilde{a}_1}} + \frac{1}{\sqrt[n-1]{\widetilde{a}_2}}\right)(\widetilde{x} - \widetilde{x}_0)^{\frac{n}{n-1}} \tag{8}$$

## 3. The Equation of Motion of the Vibro-Protected Bodies on the Rolling Bearings with Straightened Surfaces

To derive the differential equations of motion of a body, we use the equations of Ferrers, considering the Equation (8) as a holonomic link, superimposed on the vertical movement of the body. Kinetic and potential energy of the vibro-protected bodies are expressed as

$$T = M\frac{\dot{\widetilde{x}}^2 + \dot{\widetilde{y}}^2}{2}, \quad U = Mg\widetilde{y} \tag{9}$$

where $g-$ free-fall acceleration. The equation of motion of the body on the rolling bearings, in case of small oscillations, have the form:

$$\ddot{\widetilde{x}} + \frac{g}{H}\left[\frac{1}{(H\widetilde{a}_1 n)^{\frac{1}{n-1}}(\widetilde{x}-\widetilde{x}_0)^{\frac{n-2}{n-1}}} + \frac{1}{(H\widetilde{a}_2 m)^{\frac{1}{m-1}}(\widetilde{x}-\widetilde{x}_0)^{\frac{m-2}{m-1}}} - 1\right](\widetilde{x}-\widetilde{x}_0) = 0 \tag{10}$$

So, movement of the protected body (even in a case of small oscillations) is to be described by a nonlinear equation. Let us consider the case, when $n = m$, then Equation (10) will have the form

$$\ddot{\widetilde{x}} + \omega_0^2\left[\frac{\widetilde{N}_n}{\sqrt[n-1]{(\widetilde{x}-\widetilde{x}_0)^{n-2}}} - 1\right](\widetilde{x}-\widetilde{x}_0) = 0 \tag{11}$$

where

$$\widetilde{N}_n = \frac{1}{\sqrt[n-1]{nH}}\left[\frac{1}{\sqrt[n-1]{\widetilde{a}_1}} + \frac{1}{\sqrt[n-1]{\widetilde{a}_2}}\right], \quad \omega_0^2 = \frac{g}{H} \tag{12}$$

At $n \to \infty$, the Equation (11) becomes nonlinear equation of the form

$$\ddot{x}(t) - \omega_0^2 x(t) + 2\omega_0^2 \mathrm{sign}\, x(t) = -\ddot{x}_0(t), \tag{13}$$

describing oscillations of a body on the supports, having a rectangular shape.

The nonlinear Equation (6) describes the motion of the vibro-protected bodies on the rolling bearings, bounded by a parabola of higher order.

Let us introduce a new notation:

$$x = \frac{\widetilde{x}}{H}, \quad x_0 = \frac{\widetilde{x}_0}{H}, \quad t = \omega_0 \tau \tag{14}$$

The Equation (11) can be reduced to an equation in dimensionless form

$$\ddot{x} + \Phi(x-x_0) - x = -x_0(t) \tag{15}$$

where

$$\Phi(x-x_0) = N_n(x-x_0)^{\frac{1}{n-1}} \tag{16}$$

$$\widetilde{N}_n = \frac{1}{\sqrt[n-1]{nH}}\left[\frac{1}{\sqrt[n-1]{a_1}} + \frac{1}{\sqrt[n-1]{a_2}}\right], \quad a_1 = \widetilde{a}_1 H^{n-1}, \quad a_2 = \widetilde{a}_2 H^{n-1} \tag{17}$$

## 4. The Study of Free Oscillations of a Body on the Rolling Bearings with Straightened Surfaces

At $x_0 = 0$ the Equations (6) and (9) take the form

$$\ddot{x} + \Phi(x) - x = 0 \tag{18}$$

As per the method of restructuring [4], we represent the solution and the nonlinear term of Equation (10) as a truncated trigonometric series

$$x = \sum_{k=1}^{\nu} A_{2k-1}\sin(2k-1)\psi, \quad \Phi(x) = \sum_{k=1}^{\nu} b_{2k-1}\sin(2k-1)\psi \tag{19}$$

where $\psi = \omega(A_1)t..$

By choosing the required number of collocation points in interval $0 \leq \psi \leq 2\pi$, we obtain the system of algebraic equations

$$
\begin{aligned}
\alpha_{11}b_{11} + \alpha_{13}b_{13} + \ldots + \alpha_{1(2\nu-1)}b_{1(2\nu-1)} &= \Phi\left(\sum_{k=1}^{\nu} A_{2k-1}\alpha_{1(2k-1)}\right) = \Phi_1(A_1, A_3, \ldots, A_{2\nu-1}) \\
\alpha_{21}b_{21} + \alpha_{23}b_{23} + \ldots + \alpha_{2(2\nu-1)}b_{2(2\nu-1)} &= \Phi\left(\sum_{k=1}^{\nu} A_{2k-1}\alpha_{2(2k-1)}\right) = \Phi_2(A_1, A_3, \ldots, A_{2\nu-1}) \\
\alpha_{\nu1}b_{\nu1} + \alpha_{\nu3}b_{\nu3} + \ldots + \alpha_{\nu(2\nu-1)}b_{\nu(2\nu-1)} &= \Phi\left(\sum_{k=1}^{\nu} A_{2k-1}\alpha_{\nu(2k-1)}\right) = \Phi_\nu(A_1, A_3, \ldots, A_{2\nu-1})
\end{aligned}
\tag{20}
$$

where

$$
\alpha_{i(2k-1)} = \sin(2k-1)\psi_i, i = 1, 2, \ldots \nu; k = 1, 2, \ldots \nu,
$$

$\psi_i$ is the argument term at the collocation point $i$. Determinator of system Equation (20) is not equal to zero for arbitrarily chosen collocation points inside the period.

Permitting this system Equation (20) in relation to coefficients $b_{2k-1}$, we obtain

$$
b_{2k-1} = \sum_{i=1}^{\nu} \alpha_{i(2k-1)}^{(-1)} \Phi_i(A_1, A_3, \ldots, A_{2\nu-1})
\tag{21}
$$

Now by substituting expression Equation (19) in Equation (18) and equating the coefficients at the similar harmonics $\sin(2k-1)\psi$, we obtain $\nu$ equations

$$
-(2k-1)^2\omega^2 A_{2k-1} + b_{2k-1}(A_1, A_3, \ldots, A_{2\nu-1}) = 0, \ k = 1, 2, \ldots, \nu
\tag{22}
$$

Regard these equations as

$$
\omega^2 = \frac{b_1(A_1, A_3, \ldots, A_{2\nu-1})}{A_1}, \ A_{2k-1} = \frac{b_k(A_1, A_3, \ldots, A_{2\nu-1})}{(2k-1)^2\omega^2}, \ (k = 2, 3, \ldots, \nu).
\tag{23}
$$

Thus, the first equation expresses a value of the frequency point of self-oscillations of the non-linear system through amplitudes of harmonic solution. The following equation determines amplitudes of higher harmonics.

Now the Equation (23) is suitable for implementation of the iterative method. Solutions, received by the iteration method, in many cases come together, since expressions for amplitudes $2k-1$ for harmonics $A_{2k-1}$ are inversely proportional to the multiplier $(2k-1)^2\omega^2$.

The method of iteration is convenient to use by specifying value

$$
A_1 \neq 0; A_3 = A_5 = \ldots = A_{2\nu-1} = 0,
$$

The first Equation (23) takes the form of amplitude-frequency characteristics, and the remaining equations allow the definition of the form of self-oscillations of the system, presented by unabridged trigonometric range.

Assigning $\psi$ value to $\frac{\pi}{6}$, $\frac{\pi}{4}$, $\frac{\pi}{2}$ and being limited to the terms $k = 1, 2, 3$ in (21), we obtain the system as

$$
b_1 = \frac{N_n}{3}\left[\left(\tfrac{1}{2}A_1 + A_3 + \tfrac{1}{2}A_5\right)^{\frac{1}{n-1}} + \sqrt{3}\left(\tfrac{\sqrt{3}}{2}A_1 - \tfrac{\sqrt{3}}{2}A_5\right)^{\frac{1}{n-1}} + (A_1 - A_3 + A_5)^{\frac{1}{n-1}}\right],
$$

$$
b_3 = \frac{N_n}{3}\left[2\left(\tfrac{1}{2}A_1 + A_3 + \tfrac{1}{2}A_5\right)^{\frac{1}{n-1}} - (A_1 - A_3 + A_5)^{\frac{1}{n-1}}\right],
$$

$$
b_5 = \frac{N_n}{3}\left[\left(\tfrac{1}{2}A_1 + A_3 + \tfrac{1}{2}A_5\right)^{\frac{1}{n-1}} - \sqrt{3}\left(\tfrac{\sqrt{3}}{2}A_1 - \tfrac{\sqrt{3}}{2}A_5\right)^{\frac{1}{n-1}} + (A_1 - A_3 + A_5)^{\frac{1}{n-1}}\right].
$$

For the first approximation, let us assume that $A_1 \neq 0, A_3 = 0, A_5 = 0$, we obtain

$$b_1 = N_n K_1 A_1^{\frac{1}{n-1}}, b_3 = N_n K_3 A_1^{\frac{1}{n-1}}, b_5 = N_n K_5 A_1^{\frac{1}{n-1}}, a = 1, \tag{24}$$

where

$$K_1 = \tfrac{1}{3}\left[\frac{1}{2^{\frac{1}{n-1}}} + \sqrt{3}\left(\frac{\sqrt{3}}{2}\right)^{\frac{1}{n-1}} + 1\right] K_3 = \tfrac{1}{3}\left[2^{\frac{n-2}{n-1}} - 1\right], K_5 = \tfrac{1}{3}\left[\frac{1}{2^{\frac{1}{n-1}}} - \sqrt{3}\left(\frac{\sqrt{3}}{2}\right)^{\frac{1}{n-1}} + 1\right]$$

$$\varphi = \omega(A_1) \cdot (t + t_0).$$

Substituting Equation (24) in expression Equation (23) we define that

$$\omega^{(1)} = \sqrt{\frac{N_n K_1}{A_1^{\frac{n-2}{n-1}}} - 1}, A_3 = \frac{N_n K_1}{9\left(\omega^{(1)}\right)^2 + 1} A_1^{\frac{1}{n-1}}, A_5 = \frac{N_n K_5}{25\left(\omega^{(1)}\right)^2 + 1} A_1^{\frac{1}{n-1}}, \tag{25}$$

The amount in Formula (19) is chosen by the odd harmonics for reasons of the symmetry of the oscillating system. When holding in the sum Equation (19) the even terms (in the process of constructing a solution), the coefficients of these terms become zero.

To determine the second approximation we believe that $A_1 \neq 0, A_3 \neq 0, A_5 = 0$.

Solving the equations of motion Equation (18) in the second approximation has the form:

$$x = A_1 \sin \omega^{(2)} t + A_3^{(2)} \sin 3\omega^{(2)} t + A_2^{(2)} \sin 5\omega^{(2)} t,$$

where

$$\omega^{(2)} = \sqrt{\left(\frac{N_n K_1}{A_1^{\frac{n-2}{n-1}}} - 1\right) - \widetilde{\Omega}_n}, A_3^{(2)} = \frac{N_n \widetilde{K}_3 A_1^{\frac{1}{n-1}}}{9\left(\omega^{(2)}\right)^2 + 1}, A_5^{(2)} = \frac{N_n \widetilde{K}_5 A_1^{\frac{1}{n-1}}}{25\left(\omega^{(2)}\right)^2 + 1},$$

$$\widetilde{\Omega}_n = \frac{N_n A_1^{\frac{1}{n-1}}}{3}\left[\left(\frac{1}{2^{\frac{1}{n-1}}} + 1\right) - \left(\frac{1}{2} + \frac{N_n K_3 A_1^{-\frac{n-2}{n-1}}}{9\left(\omega^{(1)}\right)^2 + 1}\right)^{\frac{1}{n-1}} - \left(1 - \frac{N_n K_3 A^{-\frac{n-2}{n-1}}}{9\left(\omega^{(1)}\right)^2 + 1}\right)^{\frac{1}{n-1}}\right],$$

$$\widetilde{K}_5 = \tfrac{1}{3}\left[\left(\frac{1}{2} + \frac{N_n K_3 A_1^{-\frac{n-2}{n-1}}}{9\left(\omega^{(1)}\right)^2 + 1}\right)^{\frac{1}{n-1}} - \sqrt{3}\left(\frac{\sqrt{3}}{2}\right)^{\frac{1}{n-1}} + \left(1 - \frac{N_n K_3 A_1^{-\frac{n-2}{n-1}}}{9\left(\omega^{(1)}\right)^2 + 1}\right)^{\frac{1}{n-1}}\right], \tag{26}$$

$$\widetilde{K}_3 = \tfrac{1}{3}\left[2\left(\frac{1}{2} + \frac{N_n K_3 A_1^{-\frac{n-2}{n-1}}}{9\left(\omega^{(1)}\right)^2 + 1}\right)^{\frac{1}{n-1}} - \left(1 - \frac{N_n K_3 A_1^{-\frac{n-2}{n-1}}}{9\left(\omega^{(1)}\right)^2 + 1}\right)^{\frac{1}{n-1}}\right],$$

As an example, we consider the oscillations of a vibro-protected body on the rolling bearings, the supporting surfaces of which are bounded by the parabolas of the fourth and sixth degree, with the following parameter values,

$$n = 4; \widetilde{a}_1 = 6,25 \cdot 10^{-8} \, sm^{-3}; \widetilde{a}_2 = 15 \cdot 10^{-8} \, sm^{-3}; n = 6; \widetilde{a}_1 = 1,56 \cdot 10^{-12} \, sm^{-5}; \widetilde{a}_2 = 6,6 \cdot 10^{-12} \, sm^{-5};$$
$$H = 3m; \omega_0^2 = 3,26 s^{-2}; g = 9,8 m/s^2.$$

Dependence of the system frequency on the amplitude built by the trigonometric collocation method is shown in Figure 3. The dotted line, carried out in the graph, is built on the basis of the second approximation Equation (26). The solid line is the curve of the first approximation Equation (25).

The proximity of the curves gives an idea of the rate of convergence of the iterative processes. Considering the natural vibration frequencies to infinity, at the amplitude $A_1 \to 0$ for nonlinear systems, there is a «clash». Thus, for vibro-protection systems, a bearing element of which is a high-order parabola, the «clash» phenomenon is typical for small oscillations.

The oscillation frequency of the system slowly decreases with increasing amplitude. A parabola of the second order (for small oscillations) is independent of the amplitude (Figure 3).

Figure 4 shows the graphs of the solutions, obtained by analytical methods (line 1) and by quantitative integration by using the Runge-Kutta scheme (line 2).

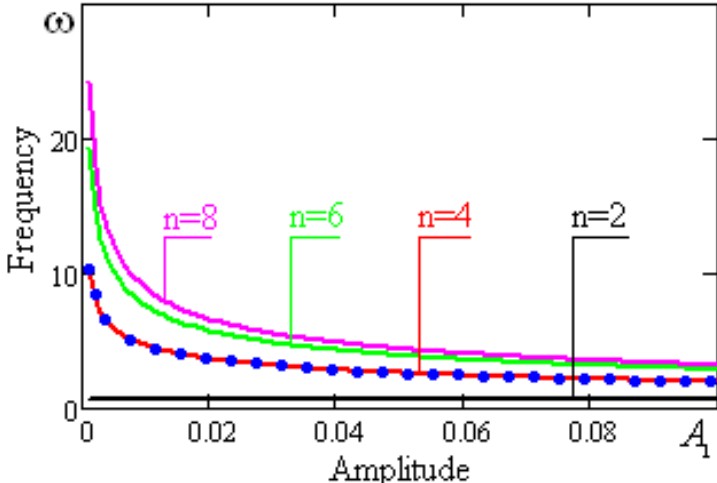

**Figure 3.** Graphs of dependence of self-oscillation oscillation frequency on various means of $n$ equation.

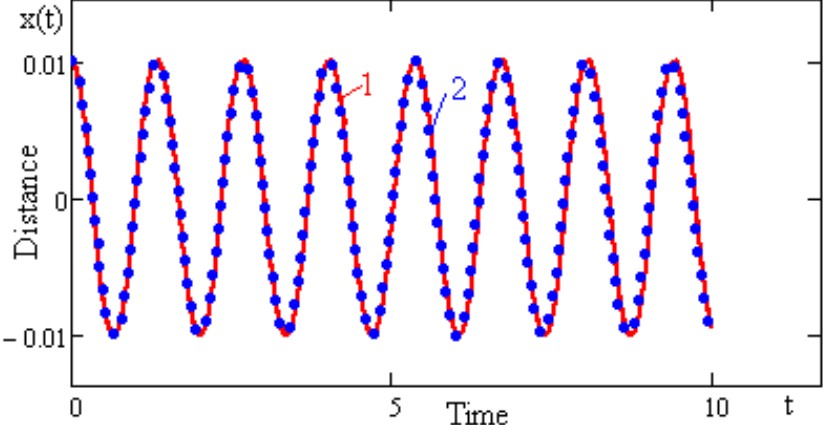

**Figure 4.** Graphs of solutions of free frequency on various means of $n$ equation.

A comparison of lines 1 and 2 shows very good exactness of the analytical calculation.

## 5. Forced Oscillations of a Vibro-Protected Body, Caused by a Movable Base

Let us study the vibrations of a body at harmonic horizontal displacement of the lower base

$$x_0(t) = Q \sin pt, \tag{27}$$

where $Q$ and $p$-dimensionless amplitude and frequency of disturbance s.

Write the solution and the nonlinear term of Equation (15) in the form

$$x = \sum_{k=1}^{v} A_{2k-1} \sin(2k-1)pt, \Phi(x - x_0) = \sum_{k=1}^{v} b_{2k-1} \sin(2k-1)pt. \tag{28}$$

Substituting Equation (28) to the equation of motion Equation (15) and limited by $k = 1, 2, 3$, we obtain a system of equations

$$-(p^2 + 1)A_1 + b_1(A_1, A_3, A_5) = -Q, -(9p^2 + 1)A_3 + b_3(A_1, A_3, A_5) = 0,$$
$$-(25p^2 + 1)A_5 + b_5(A_1, A_3A_5) = 0. \tag{29}$$

To apply the method of iteration, we present this system of equations in the form

$$A_1 = \frac{Q}{p^2+1} + \frac{b_1(A_1,A_3,A_5)}{p^2+1}, A_3 = \frac{b_3(A_1,A_3,A_5)}{9p^2+1}, A_5 = \frac{b_5(A_1,A_3,A_5)}{25p^2+1}. \tag{30}$$

The coefficients of the trigonometric series Equation (28) $b_{2k-1}$ are determined by the method of collocation. Identifying expressions Equations (16) and (28), as well as considering the relation Equations (28) and (27), we get the equation in the form

$$\sum_{k=1}^{\nu} b_{2k-1}\sin(2k-1)\varphi = N_n\Big(\sum_{k=1}^{\nu} A_{2k-1}\sin(2k-1)\varphi - Q\sin\varphi\Big)^{\frac{1}{n-1}}, \tag{31}$$

where $\varphi = pt$.

Giving to $\varphi$ values $\pi/6, \pi/3, \pi/2$, we get a system of equations relatively $b_1, b_2, b_3$ :

$$b_1 - b_3 + b_5 = N_n[(A_1-Q) - A_3 + A_5]^{\frac{1}{n-1}}, \frac{\sqrt{3}}{2}b_1 - \frac{\sqrt{3}}{2}b_5 = N_n\Big[\frac{\sqrt{3}}{2}(A_1-Q) - \frac{\sqrt{3}}{2}A_5\Big]^{\frac{1}{n-1}},$$
$$\tfrac{1}{2}b_1 + b_3 + \tfrac{1}{2}b_5 = N_n\Big[\tfrac{1}{2}(A_1-Q) + A_3 + \tfrac{1}{2}A_5\Big]^{\frac{1}{n-1}}, \tag{32}$$

from which we can find

$$b_1 = \frac{N_n}{3}\Big[\frac{1}{\sqrt{3}}\Big(\tfrac{1}{2}(A_1-Q) + A_3 + \tfrac{1}{2}A_5\Big)^{\frac{1}{n-1}} + \Big(\frac{\sqrt{3}}{2}(A_1-Q) - \frac{\sqrt{3}}{2}A_5\Big)^{\frac{1}{n-1}} +$$
$$+ \frac{1}{\sqrt{3}}\big((A_1-Q) - A_3 + A_5\big)^{\frac{1}{n-1}}\Big],$$
$$b_3 = \frac{N_n}{3}\Big[2\Big(\tfrac{1}{2}(A_1-Q) + A_3 + \tfrac{1}{2}A_5\Big)^{\frac{1}{n-1}} - \big((A_1-Q) - A_3 + A_5\big)^{\frac{1}{n-1}}\Big], \tag{33}$$
$$b_5 = \frac{N_n}{3}\Big[\Big(\tfrac{1}{2}(A_1-Q) + A_3 + \tfrac{1}{2}A_5\Big)^{\frac{1}{n-1}} - \sqrt{3}\Big(\frac{\sqrt{3}}{2}(A_1-Q) - \frac{\sqrt{3}}{2}A_5\Big)^{\frac{1}{n-1}} +$$
$$+ \big((A_1-Q) - A_3 + A_5\big)^{\frac{1}{n-1}}\Big].$$

Assuming in Equation (33) that $A_1 \neq 0, A_3 = A_5 = 0$, we obtain the values of the coefficients in the first approximation

$$b_1 = N_nK_1(A_1-Q)^{\frac{1}{n-1}}, b_3 = N_nK_3(A_1-Q)^{\frac{1}{n-1}}, b_5 = N_nK_5(A_1-Q)^{\frac{1}{n-1}}, \tag{34}$$

Taking into account Equation (34), we rewrite the relation Equation (30) in the form

$$A_1^{(1)} = \frac{1}{p^2+1}\Big[Q + N_nK_1\Big(A_1^{(1)}-Q\Big)^{\frac{1}{n-1}}\Big], A_3^{(1)} = \frac{N_nK_3}{9p^2+1}\Big(A_1^{(1)}-Q\Big)^{\frac{1}{n-1}}, A_5^{(1)} = \frac{N_nK_5}{25p^2+1}\Big(A_1^{(1)}-Q\Big)^{\frac{1}{n-1}} \tag{35}$$

Substituting Equation (35) in (28), we get the solution of Equation (15)

$$x = C_1\sin pt + \frac{N_nK_3}{9p^2+1}(A_1-Q)^{\frac{1}{n-1}}\sin 3pt + \frac{N_nK_5}{25p^2+1}(A_1-Q)^{\frac{1}{n-1}}\sin 5pt. \tag{36}$$

## 6. Results and Analysis

As an example for the parameters, given in Section 4. we have built a graph, determining the dependence of the system amplitude on the frequency of disturbances corresponding to first, third, and fifth harmonics at $Q = 3 \times 10^{-3}$ value of kinematic disturbances amplitude (See Figures 5 and 6).

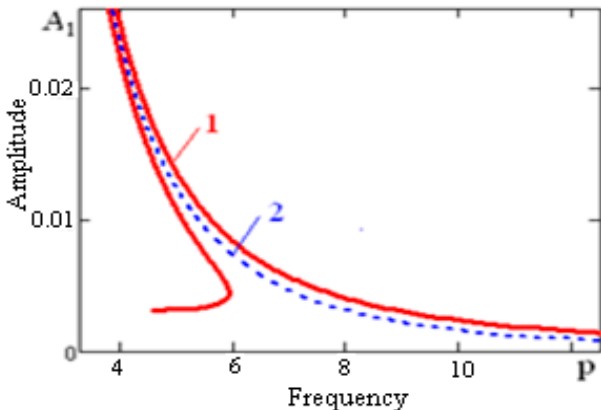

**Figure 5.** Amplitude-frequency response curve for fundamental harmonic.

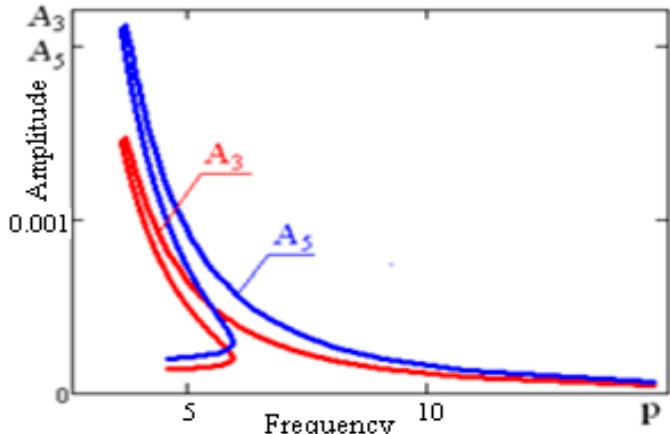

**Figure 6.** Amplitude-frequency response curve or third and fifth harmonics.

In Figure 5, line 1 corresponds to the resonance line, line 2 to the structural. In Figure 7 the resonance lines are shown corresponding to various means *n*.

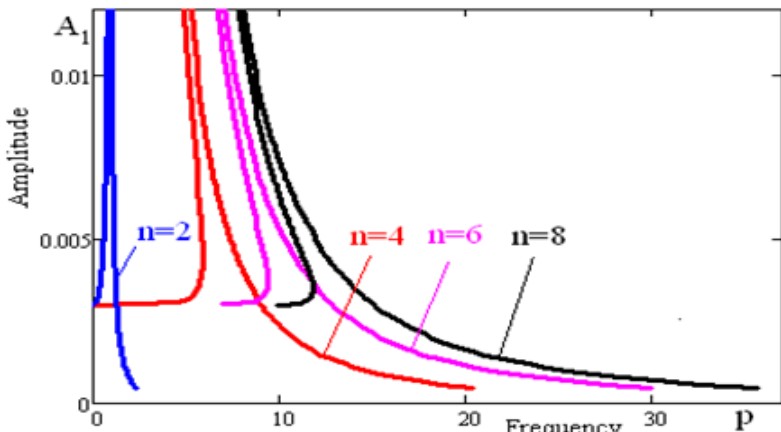

**Figure 7.** Graph of amplitude-frequency features of the main harmonic for various means of *n*.

Thus, the analysis of Figure 5 allows us to draw the following conclusions:

I. Vibratory bearings on rolling nodes, bounded by surfaces of rotation of a high order, can be attributed to nonlinear systems with soft characteristics.

II. The amplitude of the forced oscillations (up to the area of the resonant frequency) maintains a constant value, and in the resonant state, amplitude decreases to zero.

III. The calculation shows that the amplitudes of the harmonics of a higher order are smaller, as compared with the amplitudes of the fundamental harmonics (Figure 6). This demonstrates the closeness of the oscillatory process to the harmonic.

Inertia force, acting on the vibro-protected objects, is weakly dependent on the amplitude of disturbance (Figure 8). When changing the amplitude of the kinematic disturbance eight times (6–9 points), the inertia force increases in the range of 30% of the initial value.

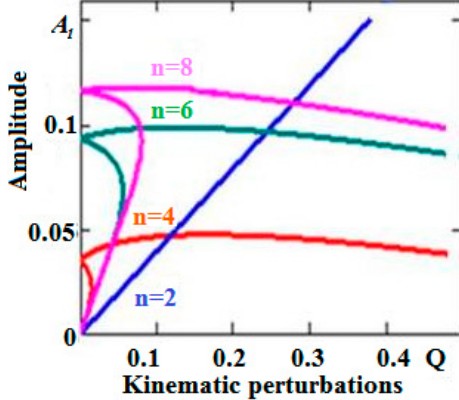

**Figure 8.** Graph of dependence of amplitude of vibro-protected body on amplitude of kinematic disturbance for various means of $n$.

For comparison we indicate that for the spherical bearings, inertia force acting on the vibro-protected body grows proportional to the amplitude of the kinematic disturbance, typical for all the linear systems. This property of the rolling bearings, bounded by the surfaces of the high order, makes them promising for creating vibration protective structures under strong kinematic disturbance.

In Figures 9 and 10, two-dimensional projections of a phase-portrait and dispensation of spectral concentration (Fourier-spectrum) of periodic oscillations of the vibro-protected bodies on the rolling bearings with straightened surfaces at $p = 5.65$ value of kinematic disturbance frequency are shown.

It is of interest to note that the range of many-fold harmonics to kinematic disturbance frequency appears in the spectrum of responses of vibro-protected systems.

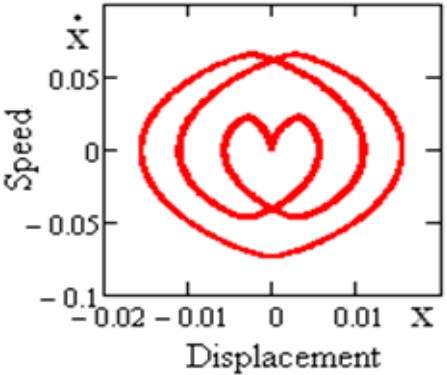

**Figure 9.** Trajectory of vibro-protected body on phase plain.

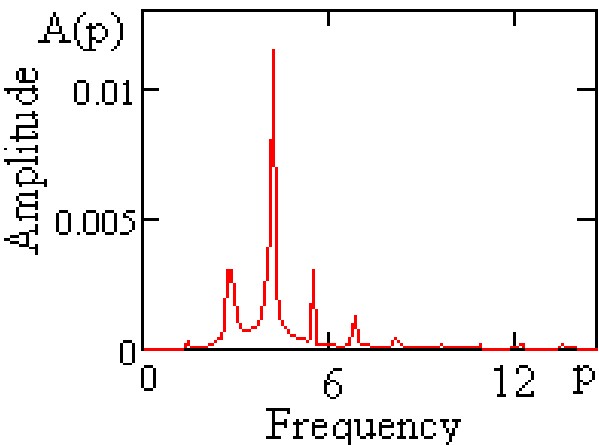

**Figure 10.** Allocation of spectral density of periodic oscillations of vibro-protected body.

## 7. Stability of Periodic Solutions

Assuming that in the case of harmonic oscillations, a component of the fundamental frequency, having the period $2\pi/p$, dominates over the higher harmonics. The periodic solution and first derivative of Equation (15) can be approximately represented as follows:

$$x = a\sin pt + b\cos pt, \dot{x} = ap\cos pt - bp\sin pt, \tag{37}$$

Let us suppose that the amplitudes $a$ and $b$ are functions of time and slowly vary depending on $t$. For the nonlinear term of Equation (15), Fourier series expansion looks as:

$$\Phi(x - x_0) = N_n C^{\frac{1}{n-1}} \sin^{\frac{1}{n-1}}(pt + \gamma) = \sum_{k=1}^{\infty} B_{2k-1}\sin(2k-1)pt + D_{2k-1}\cos(2k-1)pt \tag{38}$$

where

$$C = \sqrt{(a-Q)^2 + b^2}, tg\gamma = \frac{b}{a-Q}, B_{2k-1} = N_n K_{2k-1}\frac{(a-Q)}{\left[(a-Q)^2 + b^2\right]^{\frac{n-2}{2(n-1)}}},$$

$$D_{2k-1} = N_n K_{2k-1}\frac{b}{\left[(a-Q)^2 + b^2\right]^{\frac{n-2}{2(n-1)}}}, K_{2k-1} = \sqrt{L_{2k-1}^2 + M_{2k-1}^2}, \tag{39}$$

$$L_{2k-1} = \frac{1}{\pi}\int_0^{2\pi}\sin^{\frac{1}{n-1}}\psi\sin(2k-1)\psi d\psi, M_{2k-1} = \frac{1}{\pi}\int_0^{2\pi}\sin^{\frac{1}{n-1}}\psi\cos(2k-1)\psi d\psi, \psi = pt + \gamma.$$

Substituting Equations (37) and (38) in (15) and equating to zero the individual coefficients of the terms, containing $\sin pt$ and $\cos pt$, we have

$$\frac{da}{dt} = \frac{1}{p}\left\{(p^2 + 1) - N_n K_1\frac{1}{\left[(a-Q)^2 + b^2\right]^{\frac{n-2}{2(n-1)}}}\right\}b = X(a,b),$$

$$\frac{db}{dt} = -\frac{1}{p}\left\{\left[(p^2 + 1) - N_n K_1\frac{1}{\left[(a-Q)^2 + b^2\right]^{\frac{n-2}{2(n-1)}}}\right](a-Q) + p^2 Q\right\} = Y(a,b). \tag{40}$$

Let us consider the steady state, when amplitudes $a(t)$ and $b(t)$ in (37) are constant, i.e.,

$$\frac{da}{dt} = X(a,b) = 0, \frac{db}{dt} = Y(a,b) = 0. \tag{41}$$

In light of these conditions, from Equations (40) we can obtain that the set amplitude $a_0 = A$, $b_0 = 0$ of the periodic solution $x(t)$ is determined by the formula

$$A = \frac{1}{p^2+1}\left[N_n K_1 (A-Q)^{\frac{1}{n-1}} + Q\right]. \tag{42}$$

Let us derive the conditions for the stability of periodic solutions. We will consider small deviations $\xi$ and $\eta$ from the amplitudes $a_0$ and $b_0$, and will find out when these deviations (with increasing time) are close to zero.

From Equation (40) we get

$$\frac{d\xi}{dt} = \alpha_1\xi + \alpha_2\eta, \frac{d\eta}{dt} = \beta_1\xi + \beta_2\eta, \tag{43}$$

where

$$\alpha_1 = \frac{(n-2)}{(n-1)}\frac{1}{p}\frac{W_0}{C_0^2}(a_0-Q)b_0, \alpha_2 = \frac{1}{p}\left\{\left(p^2+1\right) - W_0 + \frac{(n-2)}{(n-1)}\frac{W_0}{C_0^2}b_0^2\right\},$$
$$\beta_1 = \frac{1}{p}\left\{-\left(p^2+1\right) + W_0 - \left(\frac{n-2}{n-1}\right)\frac{W_0}{C_0^2}(a_0-Q)^2\right\}, \beta_2 = -\frac{1}{p}\left\{\left(\frac{n-2}{n-1}\right)\frac{W_0}{C_0^2}(a_0-Q)b_0\right\}, \tag{44}$$

where

$$W_0 = \frac{N_n K_1}{C_0^{\frac{n-2}{n-1}}}, C_0 = A - Q. $$

The characteristic equation of the system has the form:

$$\lambda^2 - (\alpha_1 + \beta_2)\lambda + \alpha_1\beta_2 - \alpha_2\beta_1 = 0.a \tag{45}$$

The stability condition is given by the Routh-Hurwitz criteria, i.e.,

$$\alpha_1 + \beta_2 = 0, (\alpha_1 = 0, \beta_2 = 0).\alpha_1\beta_2 - \alpha_2\beta_1 > 0$$

to

$$\left[\left(p^2+1\right) - W_0\right]\left[\left(p^2+1\right) - \frac{W_0}{n-1}\right] > 0. \tag{46}$$

The singular point, i.e., steady system state, is a center.

The boundary of unstable periodic solutions of Equation (40) is determined by the curves.

$$p^2 = W_0 - 1, p^2 = \frac{W_0}{n-1} - 1, \tag{47}$$

and stability areas are determined by the following inequalities [15,16]

$$p^2 - (W_0 - 1) > 0, p^2 - \left(\frac{W_0}{n-1} - 1\right) > 0, p^2 - (W_0 - 1) < 0, p^2 - \left(\frac{W_0}{n-1} - 1\right) < 0. \tag{48}$$

In Figure 11 resonant curves are drawn by Equation (42). There are two branches in these graphs with respective positive and negative means of amplitude A. In Figure 11 borders of stability, built by Formula (47) are shown by lines 3; 4 and 5; 6 (lines 3; 4 for positive, lines 5; 6 for negative amplitude). From the physical perspective, positiveness of amplitude signifies phase coincidence, but negativeness of amplitudes signifies opposite-phase. Tangent lines to resonant lines are parallel to axes 7.

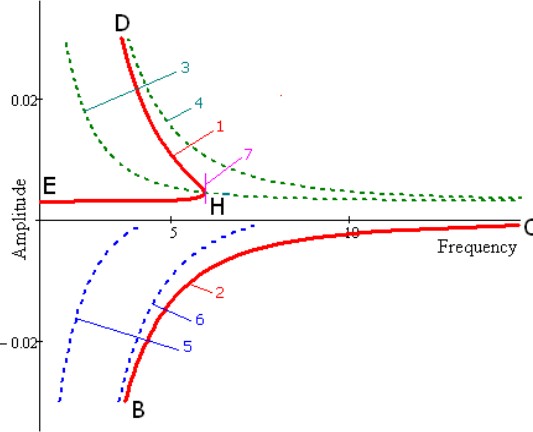

**Figure 11.** Graph of amplitude-frequency characteristics and of lines of borders of stability field.

Point H divides the upper part of the graph into stable HD and instable parts EH. The lower part of the graph is stable. Calculations were made in the following values of parameters:

$$n = 4; \ \widetilde{a}_1 = 6,25 \cdot 10^{-8} sm^{-3}; \ \widetilde{a}_2 = 15 \cdot 10^{-8} sm^{-3}; H = 3m;$$

## 8. Numerical Studies

The idea of seismic isolation of buildings in case of earthquakes with the help of rolling element linear guides is one of the simplest and most efficient ideas in the history of earthquake-proof construction [20].

The operation principle of earthquake protection devices based on rolling element linear guides is that it provides for the mobility of the building footing, which decreases the inertial force acting on it.

The seismic impacts that occur during an earthquake are classified as random actions, although in simplified calculation models that are used in practice, they are usually treated as determinate. The magnitude and nature of seismic impacts cannot be accurately predicted in advance. The instrumental records that characterize the change patterns of seismic impacts, which are characteristic of individual earthquakes, over time, never repeat one another even if they occur in the same place. Therefore, we can only discuss their similarity and, therefore, their classification in general terms.

The work [21] contain earthquake scales with their basic characteristics are listed in Table 1.

**Table 1.** Main characteristics of an earthquake.

| Intensity (Points) | Maximum Displacement Intervals, *sm* | Maximum Speed Intervals, $\frac{sm}{n}$ | Maximum Acceleration Intervals, $\frac{sm}{n^2}$ |
|---|---|---|---|
| 6 | 0.15–0.3 | 3–6 | 30–60 |
| 7 | 0.31–0.6 | 6.1–12 | 61–120 |
| 8 | 0.61–0.12 | 12.1–24 | 121–240 |
| 9 | 0.121–0.24 | 24.1–48 | 241–480 |

At a seismic level no more than 6 points, no special protection measures against earthquakes are taken. The work [21] shows that for short-period seismic disturbances ($T \leq 0.6c$), maximum acceleration values occur in ground motion with frequencies $p = 15\frac{1}{c} \div 20\frac{1}{c}$, and for long-wave disturbances-with basic frequencies $p = 6\frac{1}{c} \div 10\frac{1}{c}$. The original accelerograms provided in the paper show that the maximum values of seismic accelerations, short-period and long-period, are related to the frequencies $p = 10\frac{1}{c}$ and $20\frac{1}{c}$. Each given implementation of the ground motion process during the earthquake can be expanded into a Fourier series, taking the duration of the earthquake as the

period. If we assume that the structure responds to one of the harmonic curves in this expansion that has a frequency closest to the natural frequency of the structure, then, neglecting short-term transient processes at the beginning and the end of the earthquake, we can treat the process of the structure motion as stationary for a limited time. When seismic waves pass, both horizontal and vertical motions of the structure footing will take place. The vertical component of the acceleration usually has a somewhat smaller amplitude than the horizontal component (some 30–50%). In calculations of seismic effects, the displacement of the footing can be taken as the sum of sinusoids. To simplify the calculation, it is usually assumed that the horizontal displacement of the base is determined by the following relationship $x_0 = Q \sin pt$.

The equations of motion of a vibration-protected body on rolling element linear guides limited by surfaces of high-order rotation Equation (11) are solved numerically using Mathcad-15 software package by the Runge-Kutta method with variable step under zero initial conditions. For the numerical analysis, six rolling element linear guide models have been selected and divided into three groups. Table 2 shows the parameters of the selected rolling element linear guide models. The coefficients of the rolling element linear guide surface are calculated using the formula $y = ax^n = \frac{1}{2R^{n-1}}x^n$, where $R$ is the radius of the surface of the second order ($n = 2$). $\widetilde{a}_1$ and $\widetilde{a}_2$ are the coefficients of the lower and upper surfaces of the rolling element linear guide, respectively, $H$—Height, $n$—The order of the rolling element linear guide surface. The parameter $\widetilde{N}_n$ is calculated using the Formula (12). In the first option, different surface coefficients $\widetilde{a}_1$ and $\widetilde{a}_2$ of the rolling element linear guide were selected with constant $n$ and $H$. In the second option, different heights $H$ of the rolling element linear guide were selected with constant, $\widetilde{a}_1 \widetilde{a}_2$, and $n$. In the first option, different orders of surfaces n of the rolling element linear guide were selected with constant $\widetilde{a}_1$, $\widetilde{a}_2$, and $H$.

**Table 2.** Parameters of rolling element linear guides.

| Option | Number of Rolling Element Linear Guide Model | $R_1$ *sm* | $R_2$ *sm* | $\widetilde{a}_1$ *sm*$^{-(n-1)}$ | $\widetilde{a}_2$ *sm*$^{-(n-1)}$ | n | $H$ *sm* | $\widetilde{N}_n$ *sm*$^{\frac{n-2}{n-1}}$ |
|---|---|---|---|---|---|---|---|---|
| | 1 | 200 | 150 | $6.25 \times 10^{-8}$ | $1.481 \times 10^{-7}$ | 4 | 300 | 41.497 |
| Option 1 | 2 | 200 | 100 | $6.25 \times 10^{-8}$ | $5 \times 10^{-7}$ | 4 | 300 | 35.569 |
| | 3 | 150 | 50 | $1.481 \times 10^{-7}$ | $4 \times 10^{-6}$ | 4 | 300 | 23.713 |
| | 1 | 200 | 150 | $6.25 \times 10^{-8}$ | $1.481 \times 10^{-7}$ | 4 | 300 | 41.497 |
| Option 2 | 2 | 200 | 150 | $6.25 \times 10^{-8}$ | $1.481 \times 10^{-7}$ | 4 | 200 | 47.502 |
| | 3 | 200 | 150 | $6.25 \times 10^{-8}$ | $1.481 \times 10^{-7}$ | 4 | 100 | 59.849 |
| | 1 | 200 | 150 | $6.25 \times 10^{-8}$ | $1.481 \times 10^{-7}$ | 4 | 300 | 41.497 |
| Option 3 | 2 | 200 | 150 | $6.25 \times 10^{-8}$ | $1.481 \times 10^{-7}$ | 6 | 300 | 89.788 |
| | 3 | 200 | 150 | $3.91 \times 10^{-17}$ | $1.481 \times 10^{-7}$ | 8 | 300 | 127.112 |

To demonstrate the effectiveness of vibration isolation properties of the bearings confined by surfaces of rotation of higher order is test table, which shows the dependence of maximum acceleration vibroseismic body on rolling support on the intensity of the earthquake at various values of coefficients of surface bearings. According to the test table, selecting the parameters of the bearings can achieve a reduction of the acceleration vibroseismic body on rolling support.

Figure 12 shows the displacement, speed, and acceleration graphs of the vibration-protected body on rolling element linear guides in forced oscillation mode. In this case, the parameters of the rolling element linear guide surface in the simulation are chosen for the first model from the first option. The frequency and amplitude of the disturbance are $p = 17.4\frac{1}{c}$, $Q = 0.9\ cm$.

The main characteristics of the motion of a vibration-protected body on rolling element linear guides are as follows: maximum displacements $x_{max}$, maximum speed $\dot{x}_{max}$, and maximum accelerations $\ddot{x}_{max}$. Figure 13 shows the dependences of $x_{max}$, $\dot{x}_{max}$ on the disturbance frequency for the first model of the rolling element linear guide option. The disturbance amplitude is $Q = 0.9\ cm$. The analysis of Figure 13 makes it possible to make the following conclusion: vibration mounts on rolling bearing units limited by rotation surfaces of high-order can be regarded as non-linear systems with soft characteristics.

The maximum displacements and speeds of a vibration-protected body on rolling element linear guides have two resonant frequencies. The maximum displacement values increase slowly up to the section of resonant frequencies and decrease to zero in the resonant state.

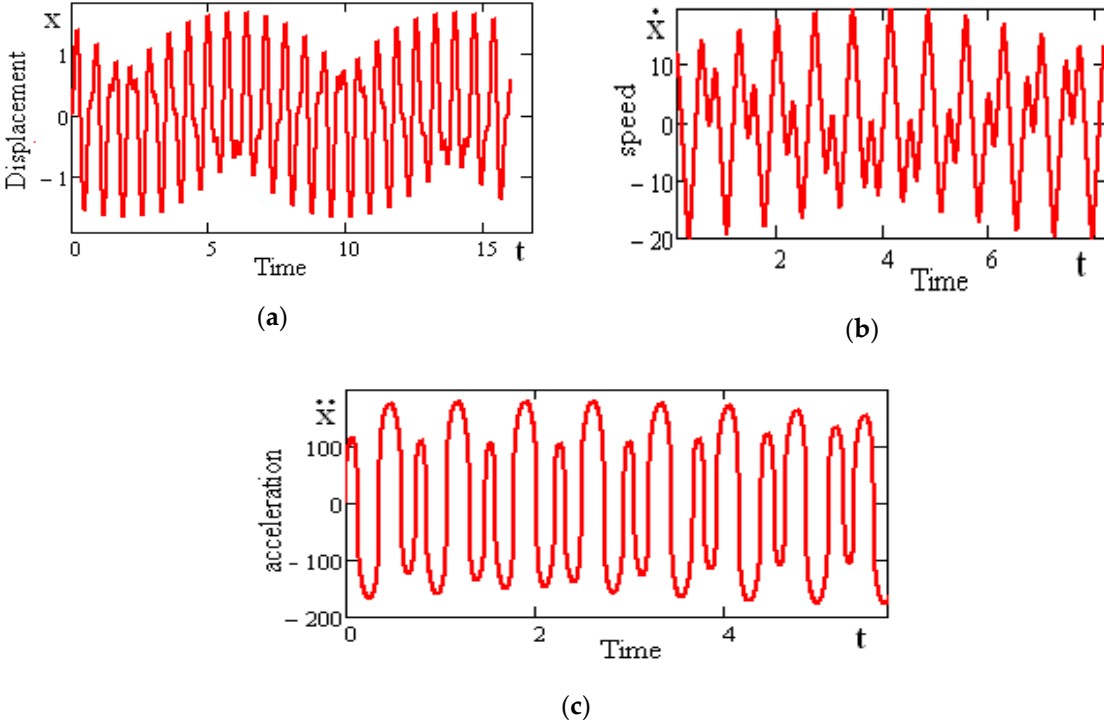

**Figure 12.** Dependence graph of: (**a**) displacement, (**b**) speed, and (**c**) acceleration of the vibration-protected body on rolling element linear guides on time.

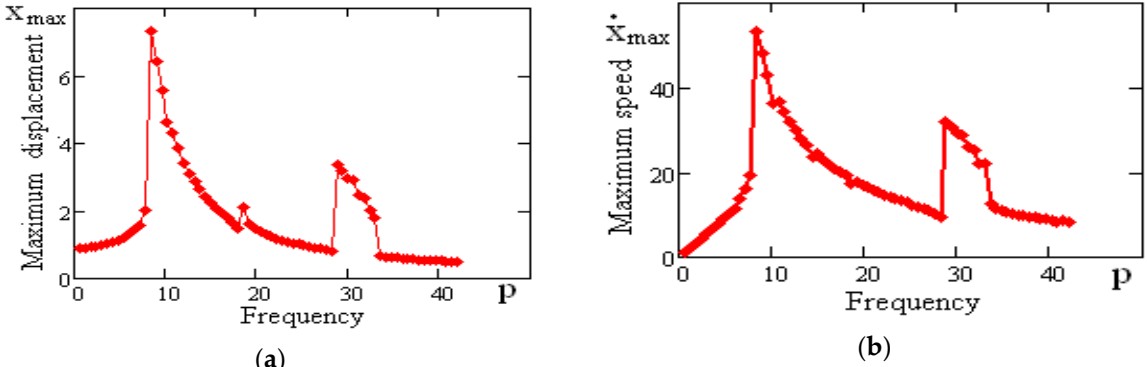

**Figure 13.** Dependence graph of maximum value of displacement (**a**) and speed (**b**) of the vibration-protected body on the rolling element linear guides on the frequency of the kinematic disturbance.

The calculation of the dynamic impacts on structures is of great importance in the design of structures. It allows determining the true load-bearing capacity of the structure more correctly. Assuming a vertical static load per rack at $10^4 kg$, calculations of the reaction forces are carried out.

Figure 14 shows the dependence of the maximum reaction force of a vibration-protected body on rolling element linear guides on the disturbance amplitude for the first rolling element linear guide model: curve-1 has been constructed using the numerical method, and curve-2-using the analytical method. Similar curves are shown in Figure 13, which gives an idea of how close the results of analytical and numerical calculations are. The frequency of the kinematic disturbances is $p = 17\frac{1}{c}$.

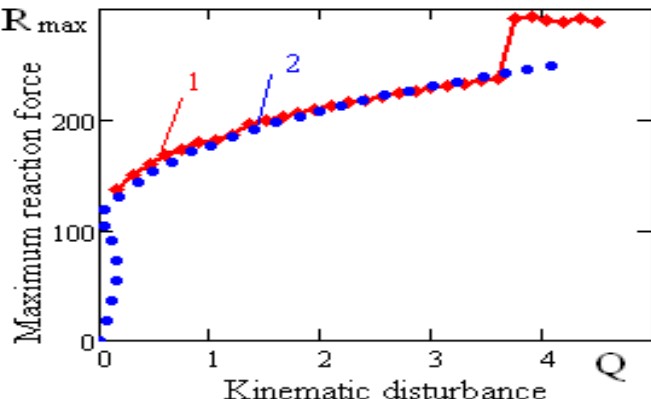

**Figure 14.** Dependence graph of maximum reaction force of the vibration-protected body on rolling element linear guides on the amplitude of the kinematic disturbance.

Figure 15 shows the dependence of the maximum reaction force of the vibration-protected body on rolling element linear guides on the frequency and amplitude of the disturbance for different values of the surface coefficients provided a constant surface order and rolling element linear guides height values (first option of Table 2). The figure shows that the maximum values of the reaction force of the vibration-protected body on rolling element linear guides decrease as the value of the rolling element linear guides surface coefficient increases. The maximum values of the reaction force of a vibration-protected body are weakly dependent on the amplitude of the kinematic disturbance. Resonance frequencies do not change significantly.

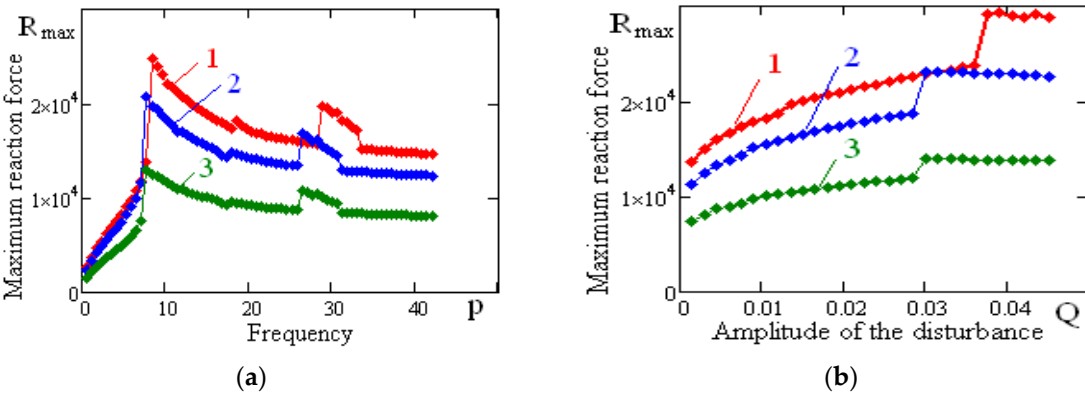

(a)　　　　　　　　　　　　　　　　　　　　　　　　(b)

**Figure 15.** Dependence graph of the maximum reaction force of the vibration-protected body on rolling element linear guides on the frequency (**a**) and amplitude (**b**) of the disturbance for different values of the coefficients of the rolling element linear guide surface.

Figure 16 shows the dependence of the maximum reaction force of the vibration-protected body on rolling element linear guides on the frequency and amplitude of the disturbance for different height values provided constant values of the coefficients and the rolling element linear guide surface order (second option of Table 2). From the figure, we can see that the maximum values of the reaction force of the vibration-protected body on rolling element linear guides increase as the value of rolling element linear guide height decreases. Abrupt changes are observed in the dependency of the maximum value of the reaction force on the amplitude of the kinematic disturbance. After the jump, the maximum values of the reaction force slowly change with the increasing value of kinematic disturbance amplitude. The resonance frequencies shift toward increasing frequency of the kinematic disturbance.

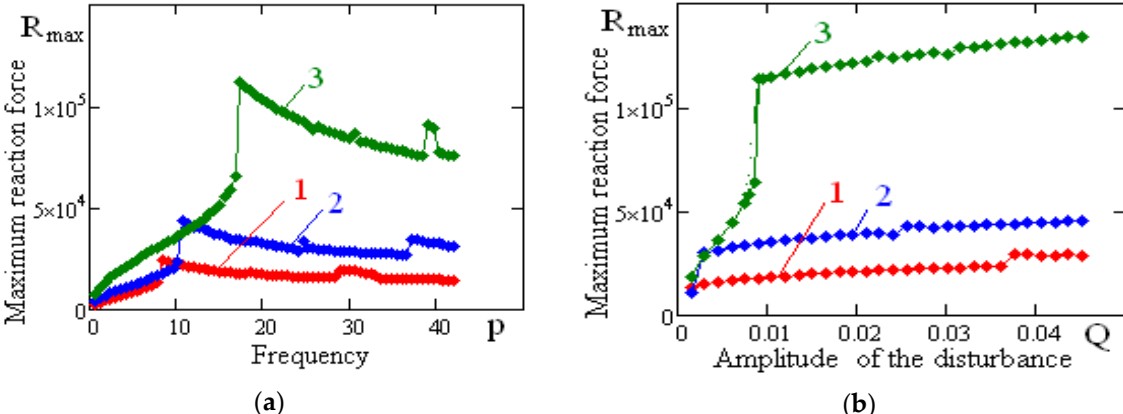

**Figure 16.** Dependence graph of the maximum reaction force of the vibration-protected body on rolling element linear guides on the frequency (**a**) and amplitude (**b**) of the disturbance for different values of rolling element linear guide height.

Figure 17 shows the dependence of the maximum reaction force of the vibration-protected body on rolling element linear guides on the frequency and amplitude of the disturbance for various values of the surface order provided constant values of the surface coefficients and the rolling element linear guide height (third option of Table 2). The maximum values of the reaction force of the vibration-protected body on rolling element linear guides increase as the order of the vibration-protected body on rolling element linear guide surface increases. The maximum values of the reaction force slowly change as the value of the amplitude of the kinematic disturbance increases. The resonance frequencies shift toward the increasing frequency of the kinematic disturbance.

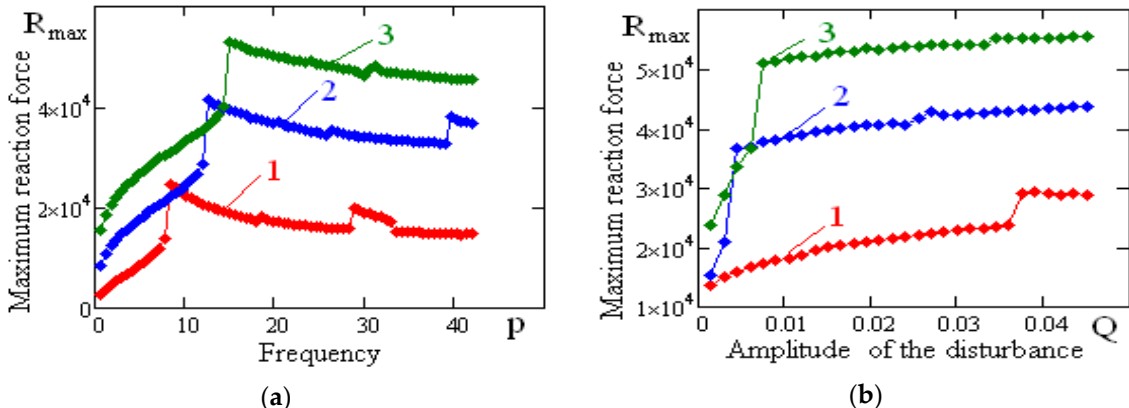

**Figure 17.** Dependence graph of the maximum reaction force of the vibration-protected body on rolling element linear guides on the frequency (**a**) and amplitude (**b**) of the disturbance for different values of surface order of the rolling element linear guide.

Thus, the analysis of the calculation allows us to draw the following conclusions: rolling element linear guides limited by high-order rotation surfaces can be regarded as non-linear systems with soft characteristics. The maximum values of the reaction force acting on the vibration-protected objects are weakly dependent on the amplitude of the kinematic disturbance. For comparison, we note that, for spherical bearings, the inertial forces acting on the vibration-protected bodies increase in proportion to the amplitude of the kinematic disturbance, which is characteristic of all linear systems.

This property of rolling element linear guides limited by high-order rotation surfaces makes them a promising solution for creating means of vibration protection for structures in the conditions of strong kinematic excitations.

## 9. Conclusions

A new mathematical model has been built and the dynamic features of vibro-protected devices, the main elements of which are rolling bearings, bounded by the surfaces of rotation of high order, have been investigated. It has been ascertained that such vibro-protected devices are highly nonlinear and a clash phenomenon appears for these systems.

In the spectrum of response emerges the range of harmonics, multiple to the frequency of kinematic disturbances. It is determined that inertial force, working on vibro-protected body on such bearings, depends little on the kinematic disturbance level.

This feature of rolling bearings, bounded by surfaces of high order, gives them potential for creation of the devices for vibro-protection of buildings under conditions of strong kinematic disturbances.

**Author Contributions:** Conceptualization, K.B. formal analysis, A.T.; investigation, K.B., A.J.; software, T.D.

**Funding:** This research was funded by Ministry of education and science of Kazakhstan grant number AP05134148.

**Conflicts of Interest:** The authors declare no conflict of interest.

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
