# Peer review of "Analysis of the Oscillating Motion of a Solid Body on Vibrating Bearers"

_machines, doi:10.3390/machines7030058_

Round 1
Reviewer 1 Report
The paper offers an interesting analysis of the oscillating motion of a solid body on vibrating bearers that may warrant publication. However, please consider the following: 1. The general English and grammar should carefully check before publication. 2. Do the authors give more another techniques compared with your technique?
Reviewer 2 Report
See file attached.

Reviewer 3 Report
This paper reported oscillating motion of a solid body on vibrating bearers, and built the basic mathematic model. It is interesting topic to the vibration absorb. The paper is integrated and concise. However, there are some suggestions after my review, which can be referred by the authors.
(1) The paper title is so long, and the author should condense it to be readable.
(2) Introduction section should be concluded at the end to highlight the contributions of this paper compared with the previous literatures.
(3) In equation (1), what is meaning of m? It is not defined in the paper.
(4)There are not any real physical experiments to support the paper. If possible, please supplement the related the experiment plan or the simplified test results. When you proposed a mathematical model, the research is far away from the final result except the experimental confirmation.
Round 2
Reviewer 2 Report
See file attached.

Reviewer 3 Report
The authors almost corrected my concerns. I am glad to accept it as its current format.
Author Response
Author are grateful to reviewer for the review work.
This manuscript is a resubmission of an earlier submission. The following is a list of the peer review reports and author responses from that submission.
Round 1
Reviewer 1 Report
The authors present a study about the oscillation of a solid body on kinematic foundations with rolling bearers bounded by the high order surfaces of rotation. The paper seems to be correct from a formal point of view but the authors are totally “self-referential” they do not compare their results with the studies of other authors or with a validated code (as a FEM code) or with experimental data. In my opinion one of these comparisons must be done.
Reviewer 2 Report
This manuscript investigates the oscillation of a solid body on kinematic foundations. The reviewer found this work not much relevant to this journal which focuses more on realistic constraints for machine design and automation design. None of these aspects were included in this submitted paper. In addition, the draft needs a lot of proof-reading as it does not convey properly the said messages of the subject.
Reviewer 3 Report
Please reformat the way the sites references are discussed in the Introduction section. Your style is based on 'This paper [ ]…. This paper [ ] ….. This paper [ ] … !'
Several missing words and spelling mistakes throughout the paper. Please revise carefully.